# CHAIN OF TIME: IN-CONTEXT PHYSICAL SIMULATION WITH IMAGE GENERATION MODELS

## ABSTRACT

We propose a novel method to improve the physical simulation ability of vision-language models. This Chain-of-Time simulation is motivated by in-context reasoning in machine learning, and mental simulation in humans. The method involves generating a series of intermediate images during a simulation. Chain of Time is used at inference time and requires no additional fine-tuning for performance benefits. We apply the Chain-of-Time method to synthetic and real-world domains, including 2-D graphics simulations and natural 3-D videos. These domains test a variety of particular physical properties, including velocity, acceleration, fluid dynamics, and conservation of momentum. We found that using Chain-of-Time simulation substantially improves the performance of state-of-the-art Image Generation Model. Beyond examining performance, we also analyze the specific states of the world simulated by an image model at each time step, which sheds light on the dynamics underlying these simulations. This analysis reveals insights that are hidden from traditional evaluations of physical reasoning, including cases where an Image Generation Model is able to simulate physical properties that unfold over time, such as velocity, gravity, and collisions domain well. Our analysis also highlights particular cases where the Image Generation Model struggles to infer particular physical parameters from input images, despite being capable of simulating relevant physical processes.

## 1 INTRODUCTION

Recent developments in Image Generation Models allow these models to generate more complex, realistic, and coherent images Chen et al. (2025); Cao et al. (2025); Liu et al. (2023); Lu et al. (2024). But despite their realism, these images often have distinct flaws, and may fail to capture real-world structures that are obvious to humans. Understanding the inner workings of Vision-Language Models (VLMs) and their internal world model representations has become a major topic in contemporary AI research (Dang et al., 2024; Chang et al., 2024; Goh et al., 2021; Bhalla et al., 2024; Zhang et al., 2024). In particular, there is a pressing question of how well VLMs and Image Generation Models represent physical properties which are required to predict how world states unfold over time. In this work, we present a method for enhancing this physical reasoning ability in Image Generation Models, which also allows us to analyze the step-by-step process that the models use to simulate physics over time.

Prior work has provided a number of tools for evaluating the physical reasoning abilities of VLMs. Comprehensive benchmarks such as PhysBench (Chow et al., 2025) and WM-ABench (Gao et al., 2025) test VLMs on a wide array of physical simulation capabilities. These evaluation benchmarks provide valuable metrics for what VLMs are capable of. However, such benchmarks do not answer the question of precisely *how* VLMs accomplish this. Our work strives to fill this gap, providing a detailed analysis of the incremental processes underlying physical simulation ability. Beyond VLMs, Meng et al. (2024) evaluates the extent to which text-to-image models - which generate images but do not take images as input - can generate images matching relational and physical constraints, using a separate VLM as an evaluator. Given the fact that the image generation models are becoming native to the vision-language models and share the world knowledge in the vison-language models (OpenAI, 2025), evaluating image generation models may also provide valuable insights into the inner workings of VLMs. Our work is, to our knowledge, unique in studying physical reasoning

Figure 1: (Left, Top) We study physical reasoning in multi-modal image generation models by providing the model a sequence of input images showing a scene in subsequent time steps, and having the model generate an image that simulates what the scene will look like some time in the future. Accurately predicting future world states requires reasoning about physical properties. (Left, Bottom) Our method, Chain of Time, allows these models to simulate a sequence of images in-context, generating one image at a time, with the last image representing the final prediction of the scene. (Right) We use four experimental domains designed to test models' ability to reason about specific physical properties: Velocity, Gravity, Fluid Dynamics, and Collision.

abilities of VLMs through Image Generation Model (IGMs), which take images and text as input and generate images as output.

In this work, we adopt a theoretical framework of mental simulation from cognitive science to understand physical reasoning and simulation abilities in Image Generation Models (Section 2). This framework helps us understand how IGMs reason about physical processes that unfold over time, by mapping input images to a latent state which is simulated with a Markov process to predict future time steps. In order to both improve physical reasoning ability of IGMs and to expose an interpretable trace of intermediate reasoning steps, we propose a novel method for in-context physical simulation, which we call Chain of Time (Section 3, Fig. 1). We test a state-of-the-art IGM with physical reasoning in four experimental domains, including two 2-D and two 3-D domains, which test four sets of physical properties: motion, gravity, fluid dynamics, and object collections. We find that Chain of Time enhances the IGM's physical reasoning abilities, enabling it to generate images which are more accurate across specific metrics over images. Further, we provide a novel analysis of the step-by-step process by which an IGM simulates the physical world, and draw key insights about what aspects of the process it succeeds and struggles with.

## 2 MENTAL SIMULATION IN HUMANS

People can reason efficiently about the physical dynamics of everyday objects. For example, if you saw a pitcher full of juice begin to fall off of a table, you might quickly and intuitively predict what sequence of events will happen next. There are many competing theories that try to explain this 'intuitive physics'. One current proposal is that people rely on a kind of 'internal physics engine' to carry out a mental simulation of a given scene (Battaglia et al., 2013; Ullman et al., 2017). While it has its critiques (see for example Ludwin-Peery et al. (2021)), this proposal finds support in cognitive science, computational modeling, cognitive development, and neuroscience (Fischer et al., 2016; Gerstenberg & Stephan, 2021; Allen et al., 2021; Fischer, 2021; Bass et al., 2021; Balaban & Ullman, 2025). More recent work suggests it is likely that humans combine various computations to carry out physical reasoning, mental simulation being just one component Hartshorne & Jing (2025); Sosa et al. (2025); Smith et al..

Given that current research suggests that step-by-step mental simulation is an important component in human physical reasoning, we adopt its formalism for studying and potentially improving upon the physical reasoning of current IGM. For our purposes here, we consider a basic version of the mental physics engine framework: Suppose that an agent observes an image $I$ that describes a

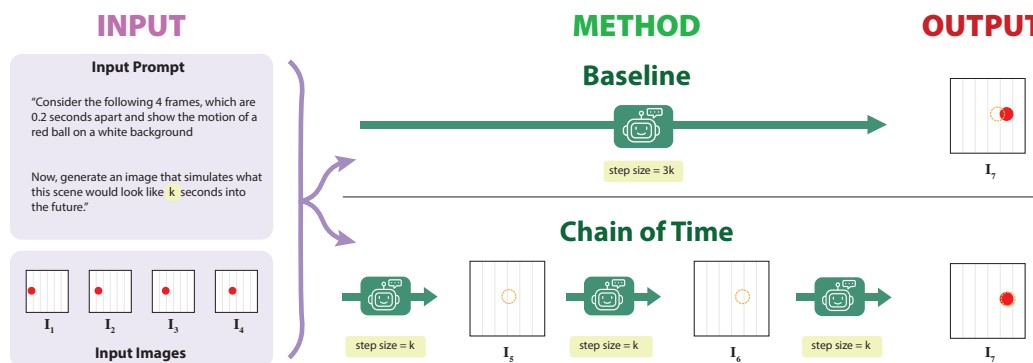

Figure 2: In our paradigm, we give a IGM a sequence of input images, along with a prompt instructing the model to simulate the scene into the future for a specified length of time (Left). As a baseline, Direct Prediction (Middle, Top) directly predicts the final state (Right). We propose a novel method, Chain-of-Time (Middle, Bottom), which instead generates a sequence of images corresponding to step-by-step simulation of the scene on the way to the predicted final state, with each mid-point image serving as input and output in mid-point computation.

scene at time $t$ in a pixel-based format, and wants to predict the state of the scene at a later time. A mental physics engine is a probabilistic transition function that can achieve this by composing three sub-functions: de-renderer $\phi$, simulator $\tau$, and renderer $\phi^{-1}$. The engine takes in the current image $I_t$, and de-renders it into the state of the world at that time, $X_t$. The engine then applies dynamic update rules to that state, corresponding to a transition $\tau$ that produces a distribution over future states of the world $X_{t+1}$. The engine may then render the state of the world back into a predicted image $I_{t+1}$.

A few notes on this overall formulation: First, while de-rendering has been studied in the context of intuitive physics in the past (e.g. Wu et al., 2017a;b; Smith et al., 2019), many other techniques exist for going from observations to physical states, and for our purposes here the specific technique is of less importance. Second, while the images $I$ are pixel-based, the underlying physical state $X$ is not, and corresponds to the 'game state' that describes in a lower-dimensional way the position, identity, and physical parameters of objects (Smith et al., 2019). Third, in computer graphics it is not strictly necessary to render the state of the world back into an image in order to answer various questions about the state, something that may hold for human mental physics as well (Balaban & Ullman, 2025).

To put it more formally, the mental simulation formalism we consider here is:

$$p(X_t \mid I_t) = \phi(I_t) + N(0, \sigma_\phi) \qquad \text{De-rendering}$$
$$p(X_{t+1} \mid X_t) = \tau(X_t) + N(0, \sigma_\tau) \qquad \text{Simulation}$$
$$p(I_{t+1} \mid X_{t+1}) = \phi^{-1}(X_{t+1}) + N(0, \sigma_{\phi^{-1}}) \qquad \text{Rendering}$$

The noise parameters $\sigma_\phi$, $\sigma_\tau$, and $\sigma_{\phi^1}$ account for perceptual noise in the de-rendering of $I$, the cognitive complexity of mental simulation of the underlying state $X$, and imperfect imagery in the rendering of the state back to an image.

Notice that the state and scene at timestep $t + 1$ depend only on the previous state and scene at timestep $t$. In other words, the formalism defines a linearly unfolding Markov Chain, that allows us to go from an initial observation $I_0$ step-by-step to a final state at time T, $X_T$, and optionally the predicted image at that time, $I_T$. While such step-by-step computations seem to underlying human mental simulation, it remains unclear whether current IGMs tasked with predicting the future state of a scene $I$ at time $T$ also go through a step-by-step simulation. Nevertheless, even if current models do not do so on their own, this framework suggests a method for bringing them more in line with human-like reasoning, which we turn to next.

## 3 CHAIN-OF-TIME SIMULATION

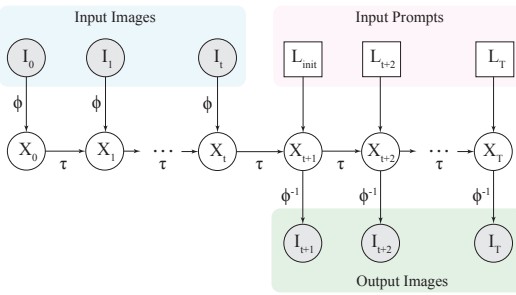

Chain-of-Time Simulation is inspired by two bodies of prior literature: the cognitive science of mental simulation (described above in Section 2) and in-context reasoning in LLMs. In-context reasoning methods with LLMs coerce a model to spell out intermediate reasoning steps in its output stream, before giving a final answer. This may be through prompting, as in Chain-of-Thought reasoning (Kojima et al., 2022), or through specialized training regimes (Guo et al., 2025; Jaech et al., 2024). These methods can significantly improve LLM performance on a variety of tasks, extending LLMs' ability to reason over complex problems with many individual steps. These reasoning chains are also distinct from traditional LLM tasks, since intermediate steps can be directly inspected my humans to interpret what exactly the model is doing. Although in some cases a model's intermediate reasoning tokens may not align with its final

Figure 3: Chain-of-Time is a composition of three components: De-rendering $\phi$, Simulation $\tau$, and Rendering $\phi^{-1}$. De-rendering operates by converting input images $I_0 \dots I_t$ into world states $X_t$, which represent a physical simulation over time. Chain-of-Time begins with an initial prompt $L_{\text{init}}$ and iteratively generates a sequence of in-context output images $I_{t+1} \dots I_T$ with follow-up prompts $L_{t+1} \dots L_T$

answer (Turpin et al., 2023), Chain of Thought reasoning has proved a valuable tool for auditing language model behavior. Various theories have been developed to try to explain precisely why and how these methods work or occasionally fail (Wang et al., 2022; Merrill & Sabharwal, 2023; Prystawski et al., 2023).

Based on these two bodies of prior literature, we propose a novel method for improving physical reasoning with in-context simulations. We call this method *Chain-of-Time Simulation* (Figure 1). We treat VLMs as a derenderer $\phi$ and a simulator $\tau$, and the IGMs as a renderer $\phi^{-1}$. The basic physical simulation task we consider is as follows: given a sequence of input images up to a given time t $I_{0:t}$, generate a new image $\hat{I_{t+k}}$ that accurately depicts what the scene will look like $k$ time steps into the future. Chain-of-Time Simulation involves two prompts (provided in Appendix B): first, a Simulation Instruction prompt that, along with a sequence of input images, instructs the model to simulate an image $k$ seconds into the future. After the IGM generates a single image, we continue with our Simulation Follow-up prompt, which instructs the model to generate another image simulated an additional $k$ seconds into the future until $t + k = T$. In our experiments, we use $T = t + 0.8$ sec and sub-steps $s \in \{0.2 \text{ sec}, 0.4 \text{ sec}\}$. As a baseline for this task, we construct a Direct Prediction Simulation prompt, which instructs the IGM to directly predict $\hat{I_{t+k}}$ given $I_{0:t}$. Note that this is equivalent to Chain-of-Time simulation with only a single timestep, i.e. $k = 0.8$ s.

### 3.1 PREVIOUS WORK

Prior work has proposed in-context reasoning methods for IGMs that use images instead of language to represent individual reasoning steps. However, our method differs from these proposals in a few critical ways. Hu et al. (2024) proposed a method to solve simple reasoning problems with a IGMs, such as geometry and spatial reasoning, and individual steps involve interleaved images and text outputs. (Xu et al., 2025) proposed a method for planning in which a IGMs generates sequential images to solve tasks such as maze navigation; their approach requires additional training. By contrast, the goal of Chain-of-Time simulation is to 1: generate the actual image, unlike the VLM that can only generate output as language; 2: improve physical simulation with IGMs, where "steps" in a chain correspond to segments of time. Further, unlike Hu et al. (2024), our method can be applied to out-of-the-box IGMs with no additional training.

## 4 EXPERIMENTS

We hypothesize that by using Chain-of-Time, IGM models will be able to achieve better accuracy than when using direct prediction. We will then use the frames created by IGMs using Chain-of-Time

simulation to reveal details about the simulation, including the initial state estimated by IGMs, the physical interaction, and the physical motion simulated by IGMs. To examine the overall validity and applicability of our method, we test Chain-of-Time on both **2D Physics** and **3D Physics**, and four domains: **2D Motion**, **2D Gravity**, **Fluids**, and **Bouncing**. We analyzed our results from three perspectives: accuracy of predicted image relative to ground truth, perceived physical interactions and motions, and perceived physical parameters.

## 4.1 EXPERIMENTAL SETUP

**Stimuli Design** As mentioned above, we used 4 different physics domains in our stimuli: `2D Motion`, `2D Gravity`, `Fluids`, and `Bouncing`. The stimuli used in 2D physics category (`2D Motion`, `2D Gravity`) were created in simulation environment, and resemble stimuli in previous studies of intuitive physics Smith & Vul (2013), Bass et al. (2021) ,Gerstenberg & Stephan (2021). The stimuli used in 3D physics category (`Fluids`) were borrowed from Wang & Ullman (2025). We manipulated the physical parameters used to generate stimuli in the 2D physics category (`2D Motion`, `2D Gravity`) and 3D physics category (`Fluids`) by changing the parameters used to simulate the stimuli. In the 3D physics category (`Bouncing`), we found real-world stimuli that have different physical parameters. For the details of the design of each stimuli, please refer to **Appendix A: Specification of the Stimuli Design**

**Experimental Procedure** We used OpenAI's GPT4-o (gpt-image-1 [1]) as the Image Generation Model model in our experiment, as of September, 2025. We also empirically tested other image generation models including DALLE-3, but found that these models were unable to simulate images of future world state with any reasonable accuracy. In order to analyze the content of generated images, we use a collection of domain-specific algorithms to identify object locations, for example x,y coordinates of generated balls, and the heights of water levels for generated fluids. These algorithms use simple tools from classic computer vision such as Hough transforms, and are further described in Appendix C.3.

At the start of each trial, the model was given 5 frames of a stimulus, showing the scene at 0, 0.2, 0.4, 0.6, and 0.8 seconds. Given these 5 frames, the model was asked to generate the a simulated frames at a time in the future, following the Initial Simulation Prompt we listed in **Appendix B: Prompt**. If Chain-of-Time 0.2s or Chain-of-Time 0.4s were used (see below), additional frames were generated following the Simulation Follow-Up Prompt we listed in **Appendix B: Prompt**.

**Sampling Details** Chain of Time generates frames at different precision, depending on a frame-rate parameter $k$. We considered two versions of Chain-of-Time with $k = 0.2$ sec and $k = 0.4$ sec. Since the final frame was 0.8 seconds into the future, 'Chain-of-Time 0.2' generated 4 frames (corresponding to 0.2, 0.4, 0.6, and 0.8 seconds after the last frame provided to the model), and 'Chain-of-Time 0.4' generated 2 frames (corresponding to 0.4 and 0.8 seconds into the future). In addition, we had a baseline termed "Direct Prediction". In this method, we asked the model to directly generate the requested final frame, 0.8 seconds into the future.

For `2D Motion`, we ran each stimulus 5 times (N=5) across all Chain-of-Time simulations and Direct Prediction. For `2D Gravity`, we ran each stimulus 20 times (N=20) across all Chain-of-Time simulations and Direct Prediction. For `Fluids`and `Bouncing`domain, we ran each stimulus 10 times (N=10) across all Chain-of-Time simulations and Direct Prediction.

**Metrics** For all domains, we considered the same analysis metrics: "Accuracy" and "Perceived Physical Motion and Physical Interaction". For accuracy, we used the Square-Root Mean-squared Error between the model's prediction and the ground truth in trials.

For the "Perceived Physical Interaction and Physical Interaction" analysis, we reveal the initial state of the physical scene perceived by the model, focusing on physical parameters we manipulated in the stimuli. We also reveal the intermediary states simulated by the model during the simulation process, and whether the intermediary states reveal the critical physical phenomena corresponding to the task by comparing them with the ground truth.

---

[1]`openai.com/index/image-generation-api/`

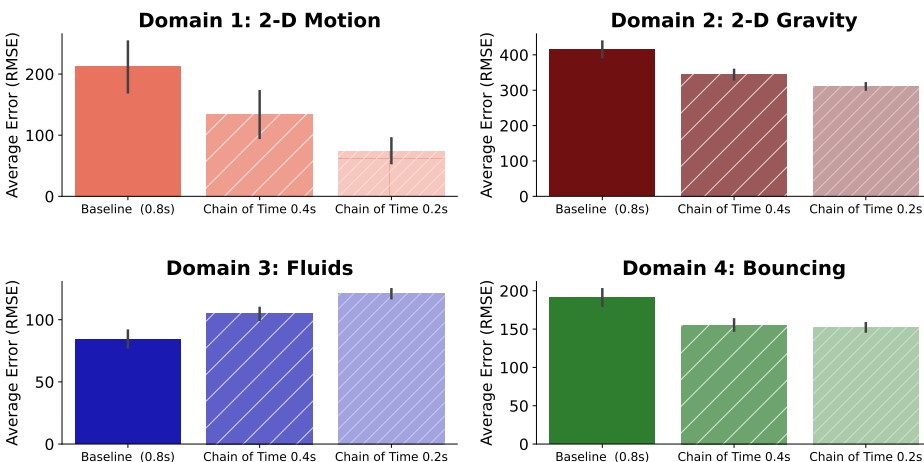

Figure 4: Prediction errors for all four domains, averaged across all data for each domain. Prediction error is measured by taking the average RMSE between the ground truth positions (location of focal object, or water level) and the positions predicted by the IGM. Error bars are 95% CI. We generally find a monotonic relationship between Chain-of-Time precision and performance. In the case of `Fluids`, we observe that the initial state simulated by the IGM is inaccurate, and this error compounds with increasing degrees of simulation, see Section 5.2.2 for detailed analysis.

## 5    RESULTS

### 5.1    ACCURACY ANALYSIS

As a reminder, we measured the IGM's accuracy in predicting the ground-truth position of the physical variable of interest (location of object, or height of water) under three different methods: Chain-of-Time 0.2s, Chain-of-Time 0.4s, and Direct Prediction. As shown in Figure 4, the finer the precision in Chain-of-Time, the better the accuracy for 3 of the 4 domains. In the `2D Motion`, Chain-of-Time 0.2s more than halves the error of Direct Prediction. The findings indicate that Chain-of-Time can increase prediction accuracy for 2D and 3D tasks.

In `Fluids` tasks, Chain-of-Time was able to enable the IGM to simulate the fluid dynamics, but due to errors in the physical parameter estimation, it failed to improve the performance. As the step-size $k$ of Chain-of-Time gets finer and finer, the error actually increases. We consider the potential cause of this increase in the following section.

### 5.2    PHYSICAL PARAMETER AND PHYSICAL MOTION ANALYSIS

Given that in the `2D Motion`, where the IMG demonstrated to ability to simulate the simplest motion which is a forward rolling motion, as the Chain-of-Time 0.2s achieved relatively Averaging Error (RMSE), we are interested in the performance of IMG in terms of simulating the complex physical interactions and motions in `2D Gravity`, `Fluids`, `Bouncing`.

Given the characteristics of Chain-of-Time, we have access to simulated images between the first timestep $t + k$ and the final timestep $T$, which are $I_{t:T}$. As described in Section 4 that we generated $I_{t:T}$ using Chain-of-Time 0.2s and Chain-of-Time 0.4s, we now used these $I_{t:T}$ to recover the estimated physical parameters perceived by the model, the estimated physical interactions, and the estimated physical motion simulated by the model. In each section, we use a single trial as an example to illustrate our point, Please refer to **Appendix C: Additional Analysis** for the same analysis on more stimuli in all four domains. In the following analysis, we focus specifically on Chain-of-Time 0.2s, as it offers the highest resolution and greatest number of images. For the analysis using Chain-of-Time 0.4s, please refer to **Appendix C: Additional Analysis** as well.

### 5.2.1 IMAGE GENERATION MODEL IS CAPABLE OF SIMULATING BOTH COMPLEX 2D AND 3D PHYSICAL MOTIONS AND INTERACTIONS

In all these three domains, we found that the IGM using Chain-of-Time 0.2s and Chain-of-Time 0.4 was generally able to create images corresponding to a simulation of the physical motions or physical interactions relevant to each domain.

In `2D Gravity`, we investigated a IGM's ability to simulate projectile motion, in which gravity causes a curved trajectory. For the purposes of illustration, we selected stimuli with speed 230, launch angle of 60 degrees, and launch position at left-bottom as an illustration. As figure 6 shows, the IGM was able to simulate the projectile motion, signified by the plot on the left that shows the curved trajectory of a projectile motion, and it was closely following the ground truth. Also, by breaking the 2-D trajectory down to the x and y components (plotted as x and y locations in two time-series in Figure 6), we found that the Image Generation Model was able to correctly simulate the interaction between the ball and gravity in Chain-of-Time 0.2s and Chain-of-Time 0.4s. The x location of the ball in the stimuli increased linearly, while the y location dropped after reaching the top due to the deceleration from gravity, which matches the characteristics of projectile motion under gravity.

In `Fluids`, we focused on whether the IGM can simulate the fluid dynamics. Although in section 5.1, we found that the average error is greater for Chain-of-time 0.2s and Chain-of-time 0.4s, we found that the model is able to simulate the fluid dynamics, indicated by the increasing of the water level as simulation proceeds. As figure 7 shows, the water level simulated by the model was increasing as time step increases. We will analyze more in section 5.2.2 about why the average error increased when Chain-of-Time simulation was used.

In `Bouncing`, we focused on the bouncing motion. Here we specifically consider stimuli with a black bouncing ball that has a medium coefficient of restitution. As Figure 5 shows, the IGM was able to simulate the bouncing motion, indicated by the y position first decreasing due to gravity, and when the ball hits the ground and started to bounce back, y positions started to increase. Notice the Figure shows a deviation between the ground truth y-positions and IGM-simulated y-positions, which analyze this further in section 5.2.2.

### 5.2.2 IMAGE GENERATION MODEL EXHIBITS PHYSICAL PARAMETER ESTIMATION ERROR FOR 3D PHYSICS SIMULATION

In Section 4, we observed that the Chain-of-Time 0.2 seconds, and Chain-of-Time 0.4 seconds had a performance drop compared to that of the Direct Prediction in the fluid dynamics domain. This

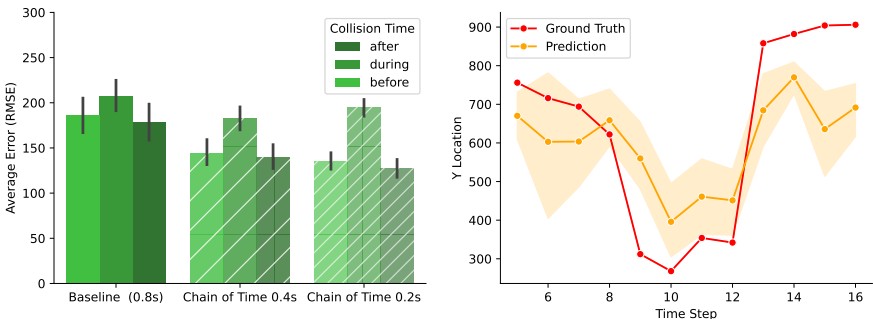

Figure 5: (Left) Prediction error rate across methods and time periods. In the collision domain, we find lower error rates in image model predictions for periods before and after the bouncing collision, compared with time periods during which the collision occurs. This disparity increases with Chain of Time, since performance improves for the before/after periods, but error remains high for the collision time period. (Right) Simulated ball location (orange) using Chain-of-Time 0.2s in the Bouncing domain follow a similar U-shaped curve as the ground truth ball location (red). Ball locations are shown here for a single video (orange), with predictions aggregated across all samples for the three time periods (before/during/after collision).

motivated us to analyze the model's perception of the initial sequence of states $X_{0:t}$, which includes physical parameters like flow rate and the initial water level in the case of `Fluids`.

We analyzed the simulated images $\hat{I_{t:T}}$ generated by the IMG in `Fluids`. More specifically, we analyzed the reported water level for stimuli with slow (25 frames / second) and fast 75 frames / second) flow rate, and low (1/12 full and 3/12 full) and high (7/12 full and 9/12 full) initial water levels. As Figure 7 shows, the IMG exhibited sensitivity towards initial water level, but no sensitivity towards the change of flow rate between the stimuli, as changing the initial water level from high to low changed the intercept of the dotted line down to the intercept of the solid line (Figure 7, left). But the flow rate did not change the slope of both dotted and solid line (Figure 7, right).

The sensitivity to initial water level means that IMG was picking up the obvious visual cue from the input images, but this non-sensitivity towards flow rate showed that IMG was estimating the flow rate with great error. Flow rate is a critical physical parameter during this task, which ensures that the glass mugs are being filled with correct amount of water at each given time. Therefore, this analysis suggests that the estimation errors about initial state $X_{0:t}$ can happen, especially for physical parameters that are more complex than the ones that can be picked up by visual cues. This estimation error led to greater accumulated error as the timestep $k$ becomes smaller. This explains why Chain-of-time is worse than Direction Prediction, and why Chain-of-time 0.2s is actually worse than Chain-of-time 0.4s in the fluid dynamics domain, and why overall Chain-of-time simulation is worse than Direct Prediction.

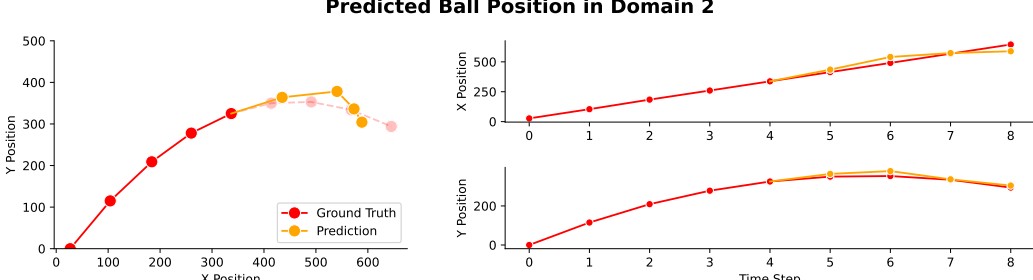

Figure 6: Predicted projectile position for `2D Gravity` over time for a single trial using Chain-of-Time 0.2s. Red represents the ground truth ball location and orange is the simulated ball location at each time step, averaged across 20 samples. (Left) Projectile location in $(x, y)$ coordinate space (Right) Predicted x-location and y-location as a function of time.

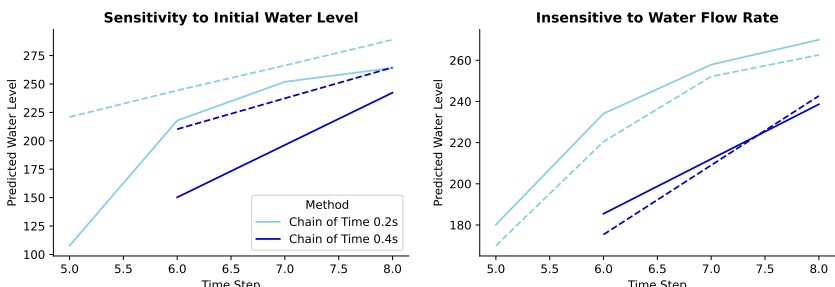

Figure 7: In the `Fluids` domain, we find that the IGMs are able to simulate water levels increasing over time. Here we show water levels in generated images steadily increasing as a function of time. (Left) The model is sensitive to initial water levels, with solid lines representing a low water level at the initial time of simulation $t$ and dotted lines representing a high initial water level. (Right) However, the model is insensitive to the flow rate of water, with water level consistently increasing at the same rate during simulations. Solid lines represent a slow flow rate and dotted lines represent a 3x faster flow rate.

Besides the flow rate parameter in `Fluids`, another complex parameter we controlled for is the coefficient of restitution in `Bouncing`. Therefore, we are interested to see the model's performance on estimating coefficient of restitution.

As mentioned in Section 5.2.1, we observed deviation between ground truth positions and the simulated positions during frame 10 and frame 16. This is the range of frames that includes the deformation (the during partition), which the ball hits the ground, deform, and bounce back. As shown in the 5, the slope starting at timestep 10 for the simulated y position is significantly smaller than that of the ground truth y position, indicating that the model think the ball started to bounce back slower than it is supposed to be. This indicates an estimation error on the coefficient of restitution.

Furthermore, as shown in Figure 5, when we expand the analysis to all the stimuli in the `Bouncing`, we see that the RMSE actually increased when the IGM is simulating the deformation partition (during), and is significantly higher than the other two partitions in Chain-of-Time 0.2 seconds and Chain-of-Time 0.4 seconds. This further reinforces the conclusion that the IGM made an estimation error in the coefficient of restitution when simulating the deformation phase.

## 6 DISCUSSION

Motivated by mental simulation in humans and in-context reasoning in Large-Language Models, we presented a method for step-by-step physical simulation in Image Generation Models. In this Chain-of-Time method, a prediction is sliced into finer precision, with mid-point frames being fed as input to the next step in an unfolding systematic process. We assessed the Chain of Time method for differing degrees of precision, across different physical domains, using different quantitative and qualitative metrics, including overall accuracy compared to ground truth, and the recovery of physical parameters.

Our results suggest that while an Image Generation Model may have some degree of physical simulation ability when paired with a VLM, accuracy degenerates significantly when simulating further into the future. Our Chain-of-Time method that inspired by the mental simulation theory in humans seems to greatly improve this ability, particularly with long simulations. When using this method, we will be able to access the simulated images produced by the image model, and probe the model's perception over critical physical parameters, physical motions, and physical interactions. We found that the model is capable of simulating both 2D and 3D physical motions and interactions accurately. But, they have various problems when simulating 2D and 3D physics, which they will slow down the simulation when 2D physics is simulated, or they will estimate the wrong physical parameters when 3D physics is simulated.

Our work is one step towards a more general method of step-by-step simulation in IGM, and many open questions and directions of research remain. For example, while we considered several settings of the precision (time-step $k$), there is a trade-off between the potential accuracy gained by better precision, and the resulting drain on computational resources. The precision that corresponds to the optimal trade-off is left for further exploration, and may depend on the target domain. In addition, we found that greater precision can *compound* error, if the initial parameters are not correctly measured or observed, and finding a way to assess this independently to know whether Chain-of-Time will be beneficial is another avenue for future work. More generally, we see great value in using Chain of Time to examine other aspects of physical reasoning not directly touched on here, including judgments of causality and non-simulation-based physical reasoning such as heuristics and abstractions. Also, while our work was inspired by research examining mental simulation in humans, our Chain of Time method and the results offers suggestions in the reverse direction for further study in people. To be specific, while much of the work on mental simulation assumes people unfold a physical scene step-by-step, the exact step-size and its possible consequences is mostly left unexamined.

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

## A    STIMULI DESIGN AND SPECIFICATION OF THE STIMULI

### A.1    2D MOTION

The stimulus shows a 2D red ball being launched at the left middle of the screen on a white background. There is no gravity or friction enabled, and no visible boundaries on the white surface. The white surface is flat and featureless, allowing the red ball to travel without obstructions, as shown in Figure 1. We varied the speed at which the ball rolled over the white surface with three different speeds: 100, 300, and 500 (in units of pixels/second). This gives us This gives us 3 stimuli in total. The stimuli and the generated image from the image generation model are in the same resolution (1024*1024) natively.

### A.2    2D GRAVITY

The stimulus shows a 2D red ball being launched at various positions (left-middle, left-bottom, right-middle, right-bottom) on a white background. There is gravity enabled, and no visible boundaries on the white surface. The white surface is flat and featureless, allowing the red ball to travel without obstructions, as shown in Figure 1. We varied the speed at which the ball rolled over the white surface with three different speeds: 230, 240, and 250 (in units of pixels/second) and with three different angles: $45°$ and $60°$. This gives us 24 stimuli in total. The stimuli and the generated image from the image generation model are in the same resolution (1024*1024) natively.

### A.3    FLUIDS

The stimulus shows a 3D glass mug being filled with water on a light sky-blue background. The water is emitted from a dark-grey pipe into the glass mug, as shown in Figure 1. There is gravity enabled, and the amount of water are limited within a range that it will never overflow. We varied the flow rate: 25, 50, and 75 (in frames/second), the type of cups: small, medium, and large, and the ground truth water levels: 1/12 full, 3/12 full, 5/12 full, 7/12 full, 9/12 full. The stimuli are cropped to 1024*1024 resolution, keeping only the water pipe and the glass water mug. This will give us 45 stimuli in total. The generated images from the image generation model are in the same resolution (1024*1024) natively.

### A.4    BOUNCING

The stimulus shows a 3D object being launched from the top and bouncing back after hitting the ground as shown in Figure 1. There is gravity enabled, and the objects all have specific coefficient of restitution. We varied the coefficient of restitution of the object by including various balls made with different materials, and the falling rate. We numbered each object from 1 to 9. Ball 1 is a transparent toy bouncing ball; Ball 2 is a white bouncy ball; Ball 3 is a black bouncy ball; Ball 4 is a squash ball; Ball 5 is a tennis ball; The velocity we varied for these 5 balls includes 25 and 50 (frames / second). Ball 6 is a soccer ball; Ball 7 is a tennis ball; Ball 8 is a purple bouncy ball; Ball 9 is a tennis ball. The velocity we varied for these 5 balls includes 10 and 15 (frames / second). We will also seperate each stimuli into three partitions. The first partition is "before", which is the sequence of frames that shows the motion of the ball right before it hits the ground and starts to deform; The second partition is "during", which is the sequence of frames that shows the motion of the ball hitting the ground, deforming, and starts to bounce back; The third partition is "after", which is the sequence of frames that shows the ball bouncing back upwards. This give us 54 stimuli in total.

The resolution of the stimuli were resized to 1080*720, and the generated images from the image generation model are in 1536*1024, and were resized into 1080*720.

# B PROMPTS

For prompt, we designed three parameters for the prompts we used in four domains.

For parameter **number of seconds forward**, which is $k$, we chose from $k \in [0.2, 0.4, 0.8]$.

For parameter **direction**, we choose from leftbottom, leftmiddle, rightmiddle, and rightbottom.

For parameter **scene content**, we choose it based on the domain. In `Fluids`, we define the parameter as the following: "a glass mug being filled with water, at a constant rate". For `Bouncing`, when simulating "before" and "during" partitions, we define the parameters as the following: "a bouncy ball falling towards the ground". When simulating "after" partition, we define the parameters as the following: "a bouncy ball bouncing upward after hitting the ground".

## B.1 PROMPT FOR 2D MOTION

In the `2D Motion`, for our Chain-of-Time simulation method, as well as our Direct Prediction baseline, models are provided the following prompt, with different methods (Chain-of-Time 0.2s, 0.4s, and Direct Prediction) varying the {{number of seconds forward}} parameter:

> **Simulation Instruction Prompt**
>
> ```
> Consider the following 5 frames, which show the motion of a red
>     ball on a white background. Note that each frame is precisely .2
>      seconds apart.
>
> Now, please generate an image that simulates what this scene would
>     look like {{number of seconds forward}} Seconds into the future.
>
> Make sure that your image is 2d and consists of a single red circle
>      on a solid white background. Ensure that the circle is exactly
>     the same size as the input images. Assume that there is no
>     friction, the ground is flat, and the ball can pass through
>     objects.
>
> {{image sequence}}
> ```

For Chain-of-Time, we used the following prompt to elicit subsequent simulation steps from the IGM:

> **Simulation Follow-Up Prompt**
>
> ```
> Now, simulate additional {{number of seconds forward}} seconds into
>     the future.
> ```

## B.2 PROMPTS FOR 2D GRAVITY

In the `2D Gravity`domain, for our Chain-of-Time simulation method, as well as our Direct Prediction baseline, models are provided the following prompt, with different methods (Chain-of-Time 0.2s, 0.4s, and Direct Prediction) varying the {{direction}} and {{number of seconds forward}} parameter:

> **Simulation Instruction Prompt**
>
> ```
> Consider the following 5 frames, which show the projectile motion
>     of a red ball being launched from the {direction}. Each frame is
>      precisely 0.2 seconds apart.
>
> You will generate an image that simulates the position of the red
>     ball {number of seconds forward} seconds forward into the future
>      after the last frame.
>
> Assume that there is gravity and the red ball continues to follow
>     the projectile motion shown in the frames provided. The scene is
>      viewed from the side, so gravity pulls downward.
>
> Make sure that the generated image is 2-D, that it contains a
>     circle exactly the same size and color as the circle in the
>     frames provided, and that the background color is white.\
>
>
> {{image sequence}}
> ```

For Chain-of-Time, we use the following prompt to elicit subsequent simulation steps from the IGM:

> **Simulation Follow-Up Prompt**
>
> ```
> Generate an additional image that simulates the position of the red
>      ball {number of seconds forward} seconds forward into the
>     future after the last frame that you generated.
>
> Assume that there is gravity and the red ball continues to follow
>     the projectile motion shown in the frames provided. The scene is
>      viewed from the side, so gravity pulls downward.
>
> Make sure that the generated image is 2-D, that it contains a
>     circle exactly the same size and color as the circle in the
>     frames provided, and that the background color is white.\
> ```

### B.3 PROMPTS FOR FLUIDS AND BOUNCING

In domains `Fluids` and `Bouncing`, for our Chain-of-Time simulation method, as well as our Direct Prediction baseline, models are provided the following prompt, with different methods (Chain-of-Time 0.2s, 0.4s, and Direct Prediction) varying the {{scene content}} and {{number of seconds forward}} parameter:

Simulation Instruction Prompt

```
Consider the following sequence of 5 images, which show {
    scene_content}. Each image frame video is precisely 0.2 seconds
    after the last frame.

Please generate an image that continues this sequence, simulating
    what the scene will look like {number of seconds forward}
    seconds further into the future after the last frame. Make sure
    that the generated image is from exactly the same perspective as
     the input images and that the background color remains the same
     color.

{{image sequence}}
```

For Chain-of-Time, we use the following prompt to elicit subsequent simulation steps from the IGM:

Simulation Follow-Up Prompt

```
Continue simulating this scene {number of seconds forward} seconds
    into the future after the last frame that you generated. Make
    sure that the generated image is from exactly the same
    perspective as the input images and that the background color
    remains the same color.
```

# C   Computer Vision Algorithms Used for Object Detection in All Four Domains

## C.1   The 2D Motion and 2D Gravity Domains

The CV algorithms we used to detect the red ball in the images generated by the image models are coded by ChatGPT-5. The code description provided by the model is as follow:

**Purpose.** Detect the centroid of the largest red region in an image.

**Steps.**

1. Read the image at `frame_path` (BGR) and convert it to HSV, since hue-based thresholding is more robust than RGB for color detection.

2. Define two HSV ranges for red (red wraps around the hue wheel), covering low hue (0–10) and high hue (170–180) with saturation/value floors to avoid dark or washed-out pixels:

$$\text{lower\_red}_1 = (0, 70, 50), \quad \text{upper\_red}_1 = (10, 255, 255),$$

$$\text{lower\_red}_2 = (170, 70, 50), \quad \text{upper\_red}_2 = (180, 255, 255).$$

3. Threshold the HSV image with both intervals and OR the results to obtain a binary mask of red pixels.

4. Find external contours on the mask. If none are found, return $(\text{None}, \text{None})$.

5. Select the largest contour by area (assumes the main red target is the biggest red blob in view).

6. Compute spatial moments for that contour. If $m_{00} = 0$ (degenerate area), return $(\text{None}, \text{None})$; otherwise compute the centroid:

$$c_x = \frac{m_{10}}{m_{00}}, \qquad c_y = \frac{m_{01}}{m_{00}}.$$

7. Return $(c_x, c_y)$ as integer pixel coordinates. If no valid region exists, return $(\text{None}, \text{None})$.

**Returns.** $(\text{int}, \text{int})$ or $(\text{None}, \text{None})$: the centroid of the largest red region, or `None`/`None` when no suitable red region is found.

And figure 8 are examples of the algorithm detecting the red ball in images generated in 2D Motion, and 2D Gravity.

## C.2   The Fluids Domain

The CV algorithms we used to detect the water blob and water level in the images generated by the image generation models are coded by ChatGPT-5. The code description provided by the model is as follow:

**Detect the water "blob" inside a glass mug and report its top/bottom y.**

**Inputs**

- `img_path`: path to RGB/RGBA image.
- `y_cut`: only analyze rows $y \geq y\_cut$ (ignore the upper part of the image).
- `alpha_thresh`: pixels with alpha $\leq$ this are treated as transparent (ignored).
- `hsv_lower` / `hsv_upper`: HSV thresholds for sky-blue liquid (OpenCV $H \in [0, 179]$).
- `erode_iters`, `close_kernel`: morphology parameters to suppress thin rims/stream and fill gaps.
- `coverage_main` / `coverage_fallback`: minimum horizontal coverage (fraction of bbox width) required for a row to be considered liquid; a fallback is used if the main threshold yields none.

- `ignore_top_rows`: discard the first $N$ rows inside the ROI to avoid picking the crop boundary.

**Steps**

1. Read the image with `cv2.IMREAD_UNCHANGED` so the alpha channel (if present) is preserved.
2. Build an alpha mask: alpha_mask $= (\alpha >$ alpha_thresh$)$. If no alpha channel exists, use an all-ones mask (all pixels opaque).
3. Crop to the region of interest (ROI): rows $y \geq y\_cut$. Crop the alpha mask the same way.
4. Convert the ROI to HSV and threshold with $[$`hsv_lower, hsv_upper`$]$ to obtain a binary color mask for the sky-blue liquid.
5. AND the color mask with the alpha mask to suppress fully/mostly transparent background.
6. Morphological cleanup:
   - Erode (`erode_iters` times) to remove thin bright rims and the pouring stream.

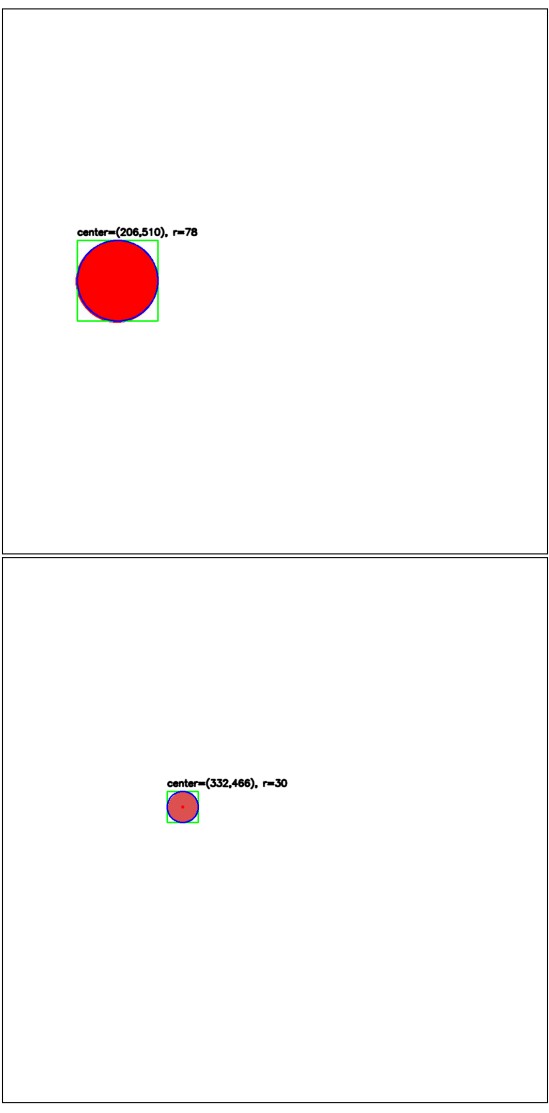

Figure 8: Examples of the algorithm detecting the red ball for stimuli used in `2D Motion`, and `2D Gravity`.

- Close (kernel size = close_kernel) to reconnect the eroded blob and fill small holes.

7. Find connected components (external contours) and select the largest-area component as the liquid blob.

8. Compute the blob's bounding box $(x, y, w, h)$ in ROI coordinates. Inside this box:
    - For each row, count foreground pixels (row_counts).
    - Mark rows as liquid if row_counts $\geq$ coverage_main $\times w$.
    - If none qualify, relax to coverage_fallback $\times w$.
    - Discard candidate rows whose index $<$ ignore_top_rows.

9. Determine water-level bounds:
    - Top $y$ (global) $= y\_cut + y + \min(\text{valid\_row\_indices})$ if valid rows exist; otherwise $y\_cut + \min(\text{contour\_y})$.
    - Bottom $y$ (global) $= y\_cut + y + \max(\text{valid\_row\_indices})$ if valid rows exist; otherwise $y\_cut + \max(\text{contour\_y})$.

10. Optional visualization:
    - Red rectangle: blob bounding box.
    - Green line: top $y$.
    - Blue line: bottom $y$.

**Notes**

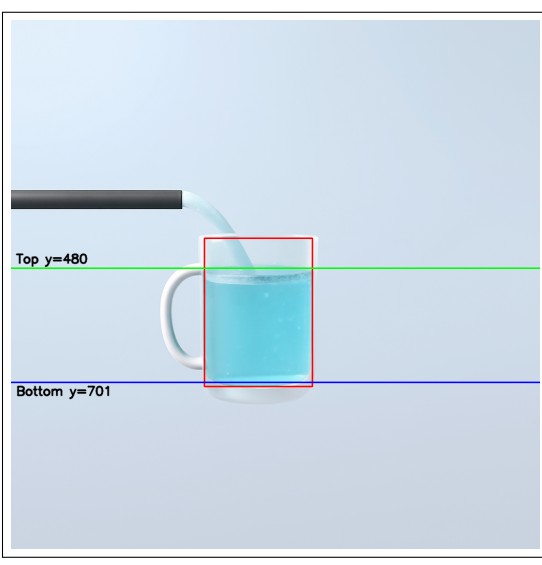

Figure 9: Examples of the algorithm detecting the water level for stimuli used in Fluids.

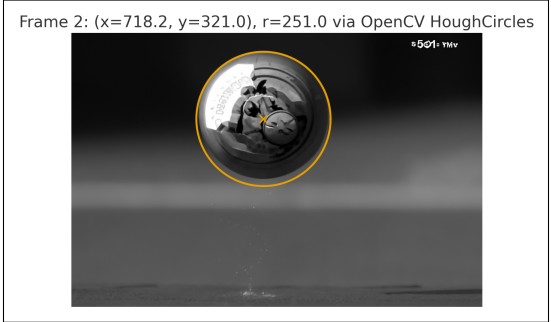

Figure 10: Examples of the algorithm detecting the ball used in Bouncing.

- HSV bounds are intentionally broad; tune to lighting and hue variations.

- Erosion suppresses 1–2 px rims and stream artifacts that otherwise bias the "top" level.

- The row-coverage rule prefers rows where a substantial horizontal span is filled, making the estimate robust to foam/splash highlights.

- Complexity is approximately $O(HW)$ per frame (thresholding + morphology + a single bbox scan).

And figure 9 is an example of the algorithm detecting the water level in images generated in `Fluids`.

## C.3 THE BOUNCING DOMAIN

The CV algorithms we used to detect the ball in the images generated by the image models are coded by ChatGPT-5. The code description provided by the model is as follow:

We detect a single ball in an RGB image using a two-stage strategy: (i) a Hough transform for circles on a contrast-enhanced, downscaled grayscale image; and (ii) a fallback based on contour circularity if no reliable Hough detection is found. The routine computes both a bounding box and a center, clamps the box to image bounds, and returns the center.

**Inputs.** An image $\mathbf{I} \in \mathbb{R}^{H \times W \times 3}$ (BGR order as in OpenCV).

**Outputs.** *Center $(c_x, c_y)$ in original-image pixel coordinates. (Note: the code also computes a bounding box $(x, y, w, h)$ but, as written, returns only the center.)*

**Preprocessing.**

1. Downscale $\mathbf{I}$ by $2\times$ (area interpolation) to reduce noise and speed up detection.

2. Convert to grayscale and apply CLAHE (clip limit = 2.0, tile size $8 \times 8$) for local contrast amplification, followed by Gaussian blur (kernel $7 \times 7$, $\sigma \approx 1.5$) to suppress noise.

**Stage 1: Hough circle detection.**

- Apply `HoughCircles` (gradient method) with parameters:

$$\text{dp} = 1.2, \quad \text{minDist} = 50, \quad \text{param1} = 100, \quad \text{param2} = 18, \quad r \in [15, 200].$$

- If one or more circles are found, choose the one with the strongest edge response. For each candidate $(\tilde{c}_x, \tilde{c}_y, \tilde{r})$, compute a thin ring mask and measure the mean Canny edge magnitude within the ring; select the circle with the maximal mean.

- Rescale $(\tilde{c}_x, \tilde{c}_y, \tilde{r})$ by factor 2 back to original resolution: $(c_x, c_y, r) = (2\tilde{c}_x, 2\tilde{c}_y, 2\tilde{r})$.

- Define a square bounding box centered at $(c_x, c_y)$ with side length $2r$: $(x, y, w, h) = (c_x - r, \ c_y - r, \ 2r, \ 2r)$.

**Stage 2 (fallback): contour circularity.**

1. If Hough detection fails, convert the original image to grayscale, blur (Gaussian $7 \times 7$), and compute Canny edges.

2. Morphological close with a $5 \times 5$ kernel to connect fragmented edges.

3. Extract external contours and filter:

   - Reject small contours: area $A < 300$ px.
   - Compute perimeter $P$ and circularity

$$\mathcal{C} = \frac{4\pi A}{P^2 + \epsilon}, \quad \epsilon = 10^{-6};$$

   keep contours with $\mathcal{C} \geq 0.7$.

4. For each remaining contour, compute the axis-aligned bounding box $(x, y, w, h)$ and score it by $\mathcal{C} \cdot A$; take the maximum-scoring contour as the ball.

5. Set center $(c_x, c_y) = (x + \lfloor w/2 \rfloor, \; y + \lfloor h/2 \rfloor)$ and approximate radius $r = \lfloor \max(w, h)/2 \rfloor$.

**Post-processing.** Clamp $(x, y, w, h)$ to image bounds: $x \leftarrow \max(0, x)$, $y \leftarrow \max(0, y)$, $w \leftarrow \min(w, W - x)$, $h \leftarrow \min(h, H - y)$. Return the integer center $(c_x, c_y)$.

**Notes and implementation details.**

- Contrast-limited histogram equalization (CLAHE) improves robustness to faint or low-contrast balls.

- The edge-strength tie-breaker favors circles with sharper boundaries rather than merely high accumulator votes.

- The circularity threshold $\mathcal{C} \geq 0.7$ trades off recall vs. precision; higher values reject more elongated shapes.

- If no suitable contour is found, the routine raises an exception (``Ball not found'').

- *Docstring mismatch:* the docstring claims to return both bounding box and center, but the function currently returns only the center; adjust as needed.

And figure 10 is an example of the algorithm detecting the ball in images generated in `Bouncing`.

# D ADDITIONAL ANALYSIS

## D.1 THE 2D GRAVITY DOMAIN

Here are the additional analysis for stimuli with speed 230 pixels/second, angle $60°$, and launching position left-bottom. We present the result generated by Chain-of-time 0.4s on the same stimuli. We can see that we observed the deceleration on the y-axis due to gravity, and we can see the projectile motion like curve on the left plot of the figure 11, matching the conclusion we reached in section 5.2.2.

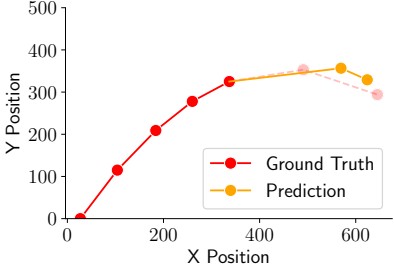 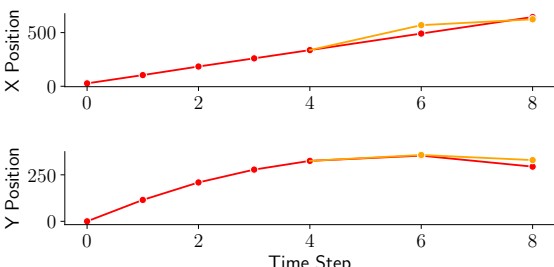

Figure 11: Predicted projectile position for 2D Gravity over time for a single trial using Chain-of-Time 0.4s. Red represents the ground truth ball location and orange is the simulated ball location at each time step, averaged across 20 samples. (Left) Projectile location in $(x, y)$ coordinate space (Right) Predicted x-location and y-location as a function of time.

Here are the additional analysis for stimuli with speed 240 pixels/second, angle $60°$, and launching position left-bottom. We present the result generated by Chain-of-Time 0.2s on the same stimuli. We can see that we observed the deceleration on the y-axis due to gravity, and we can see the projectile motion like curve on the left plot of the figure 12, matching the conclusion we reached in section 5.2.2.

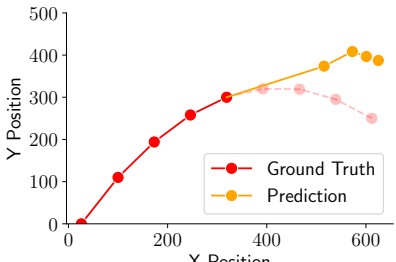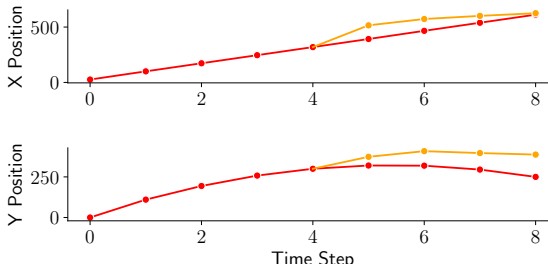

Figure 12: Predicted projectile position for 2D Gravity over time for a single trial using Chain-of-Time 0.2s. Red represents the ground truth ball location and orange is the simulated ball location at each time step, averaged across 20 samples. (Left) Projectile location in $(x, y)$ coordinate space (Right) Predicted x-location and y-location as a function of time.

Here are the additional analysis for stimuli with speed 240 pixels/second, angle $60°$, and launching position left-bottom. We present the result generated by Chain-of-Time 0.4s on the same stimuli. We can see that we observed the deceleration on the y-axis due to gravity, and we can see the projectile motion like curve on the left plot of the figure 13, matching the conclusion we reached in section 5.2.2.

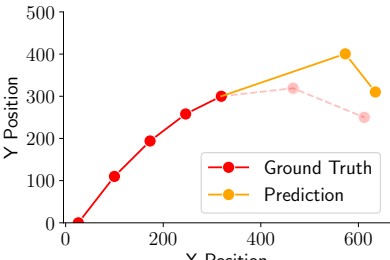 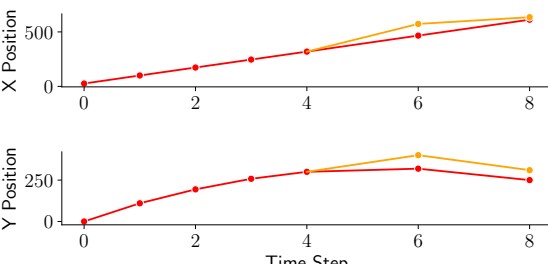

Figure 13: Predicted projectile position for `2D Gravity` over time for a single trial using Chain-of-Time 0.2s. Red represents the ground truth ball location and orange is the simulated ball location at each time step, averaged across 20 samples. (Left) Projectile location in $(x, y)$ coordinate space (Right) Predicted x-location and y-location as a function of time.

Here are the additional analysis for stimuli with speed 220 pixels/second, angle $60°$, and launching position right-middle. We present the result generated by Chain-of-Time 0.2s on the same stimuli. We can see that we observed the deceleration on the y-axis due to gravity, and we can see the projectile motion like curve on the left plot of the figure 14, matching the conclusion we reached in section 5.2.2.

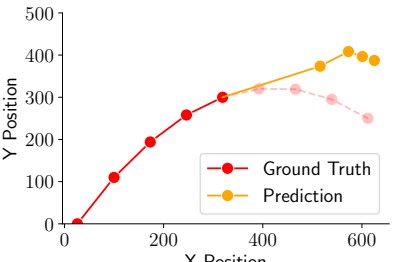 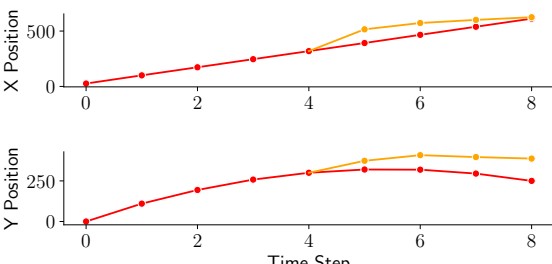

Figure 14: Predicted projectile position for `2D Gravity` over time for a single trial using Chain-of-Time 0.2s. Red represents the ground truth ball location and orange is the simulated ball location at each time step, averaged across 20 samples. (Left) Projectile location in $(x, y)$ coordinate space (Right) Predicted x-location and y-location as a function of time.

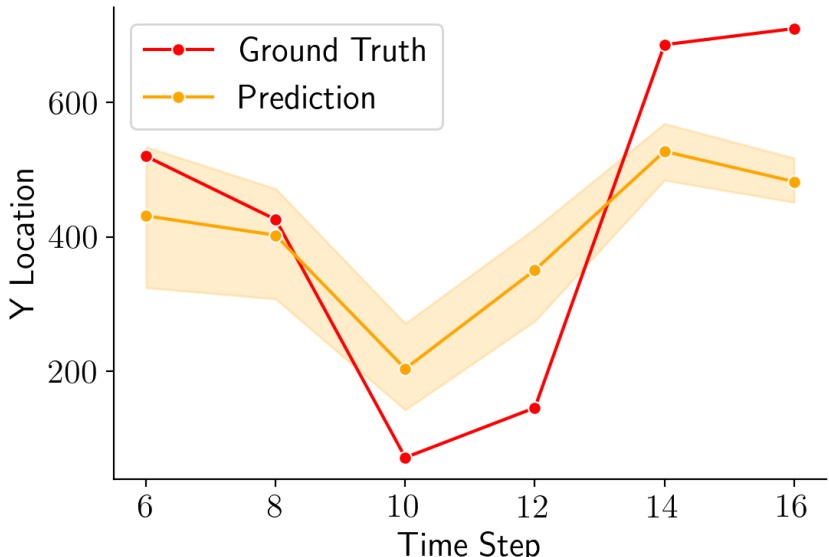

Figure 15: Simulated ball location (orange) using Chain-of-Time 0.4s in the Bouncing domain follow a similar U-shaped curve as the ground truth ball location (red). Ball locations are shown here for a single video (orange), with predictions aggregated across all samples for the three time periods (before/during/after collision).

## D.2  THE BOUNCING DOMAIN

Here are the additional analysis on data generated by Chain-of-Time 0.2s and 0.4s for ball 2 at velocity 50 frames/second. This is the result generated by Chain-of-Time 0.4s simulation. We can see that the bouncing motion is shown by the deep V shaped curved, and the IGM underestimated the coefficient of restitution, since IGM thinks the ball is bouncing back slower, which the conclusion matches with the conclusion we reached in section 5.2.2

Here are the additional analysis on data generated by Chain-of-Time 0.2s and 0.4s for ball 4 at velocity 50 frames/second. We can see that the bouncing motion is shown by the deep V shaped curved, and the IGM underestimated the coefficient of restitution in Chain-of-Time 0.4s, since IGM thinks the ball is bouncing back slower.

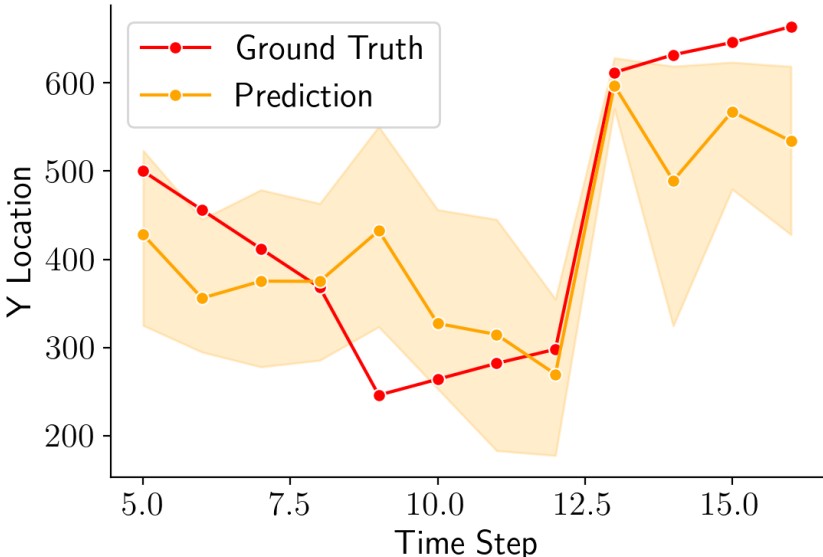

Figure 16: Simulated ball location (orange) using Chain-of-Time 0.2s in the Bouncing domain follow a similar U-shaped curve as the ground truth ball location (red). Ball locations are shown here for a single video (orange), with predictions aggregated across all samples for the three time periods (before/during/after collision).

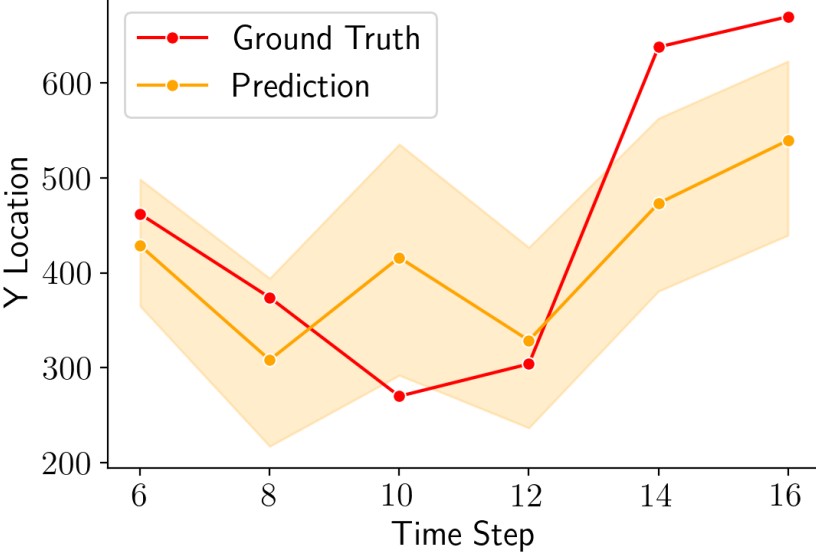

Figure 17: Simulated ball location (orange) using Chain-of-Time 0.4s in the Bouncing domain follow a similar U-shaped curve as the ground truth ball location (red). Ball locations are shown here for a single video (orange), with predictions aggregated across all samples for the three time periods (before/during/after collision).

Here are the additional analysis on data generated by Chain-of-Time 0.2s and 0.4s for ball 7 at velocity 15 frames/second. We can see that the bouncing motion is shown by the deep V shaped curved, and the IGM underestimated the coefficient of restitution, since IGM thinks the ball is bouncing back slower.

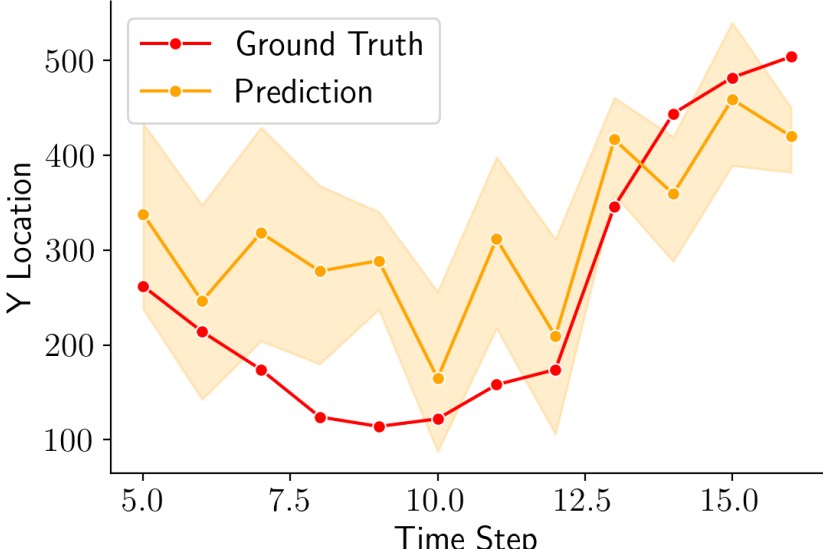

Figure 18: Simulated ball location (orange) using Chain-of-Time 0.2s in the Bouncing domain follow a similar U-shaped curve as the ground truth ball location (red). Ball locations are shown here for a single video (orange), with predictions aggregated across all samples for the three time periods (before/during/after collision).

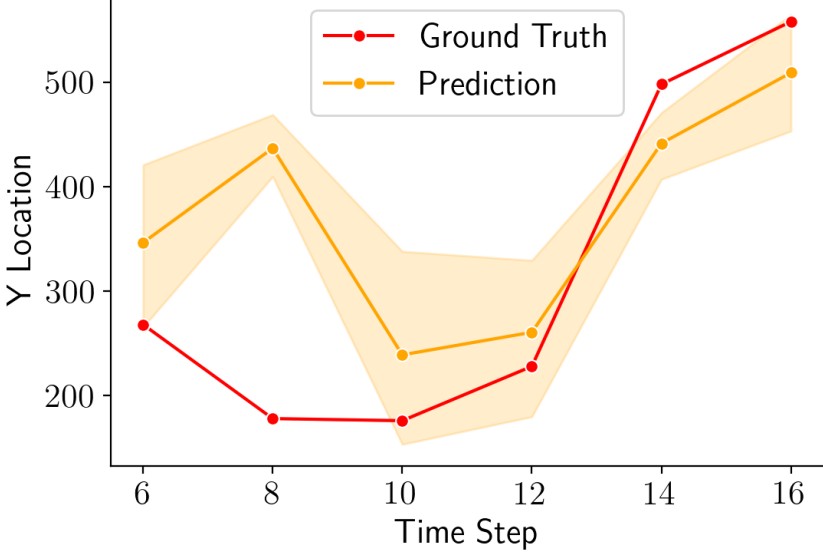

Figure 19: Simulated ball location (orange) using Chain-of-Time 0.4s in the Bouncing domain follow a similar U-shaped curve as the ground truth ball location (red). Ball locations are shown here for a single video (orange), with predictions aggregated across all samples for the three time periods (before/during/after collision).

Here are the additional analysis on data generated by Chain-of-Time 0.2s and 0.4s for ball 9 at velocity 10 frames/second. We can see that the bouncing motion is shown by the deep V shaped curved, and the IGM underestimated the coefficient of restitution, since IGM thinks the ball is bouncing back slower.

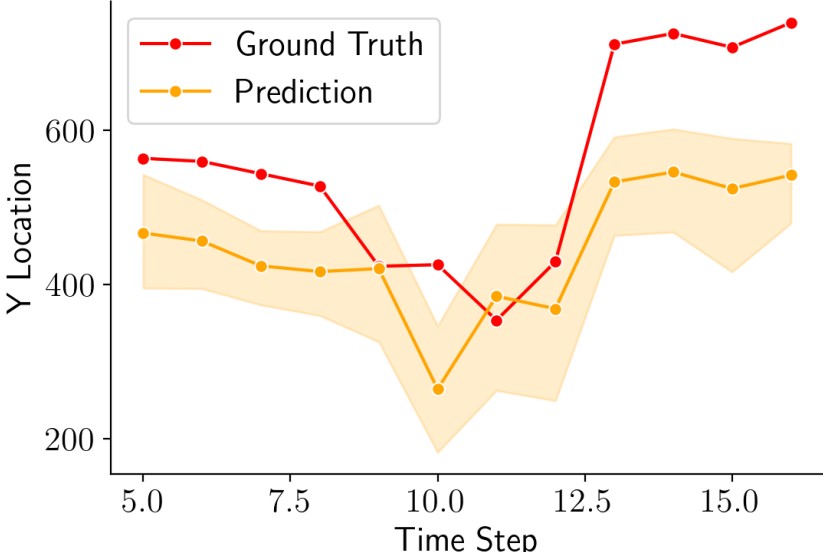

Figure 20: Simulated ball location (orange) using Chain-of-Time 0.2s in the Bouncing domain follow a similar U-shaped curve as the ground truth ball location (red). Ball locations are shown here for a single video (orange), with predictions aggregated across all samples for the three time periods (before/during/after collision).

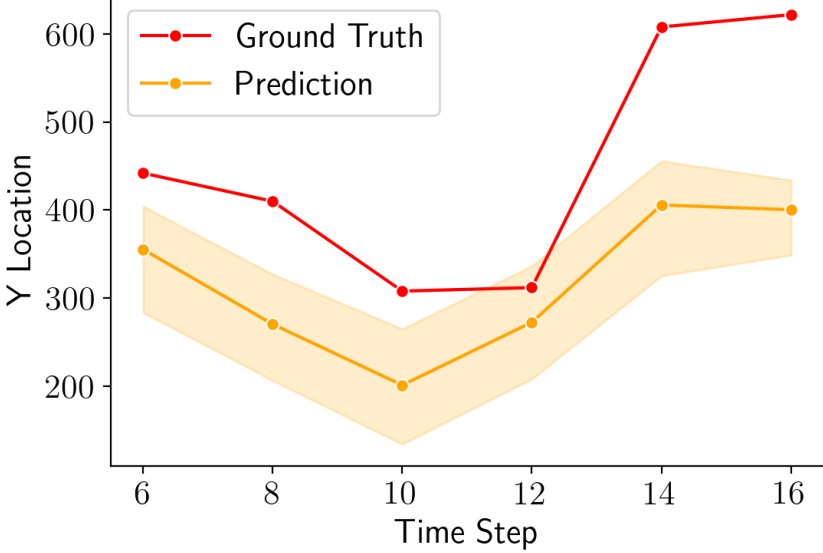

Figure 21: Simulated ball location (orange) using Chain-of-Time 0.4s in the Bouncing domain follow a similar U-shaped curve as the ground truth ball location (red). Ball locations are shown here for a single video (orange), with predictions aggregated across all samples for the three time periods (before/during/after collision).

# E    LANGUAGE MODEL STATEMENT

LLMs were used in this work for literature review and for coding assistance with constructing computer vision algorithms described in Appendix C.3

