# OpenReview forum: "Chain of Time: In-Context Physical Simulation with Image Generation Models"
_ICLR.cc/2026/Conference — Submitted to ICLR 2026_

### Official Review · Reviewer_7Pnk · 2025-10-30

**Soundness:** 2
**Presentation:** 2
**Contribution:** 2
**Rating:** 4
**Confidence:** 4

**Summary:**

This paper proposes Chain-of-Time, a method for improving physical simulation in Image Generation Models by generating intermediate frames step-by-step, inspired by mental simulation in humans and chain-of-thought reasoning in LLMs. The authors test on four domains (2D Motion, 2D Gravity, Fluids, Bouncing) and show improvements over direct prediction in most cases.

**Strengths:**

Moderately Novel approach: The connection between human mental simulation theory and in-context reasoning for IGMs is creative and well-motivated.

Interpretability: The method provides interpretable intermediate steps that reveal the model's physical reasoning process.

Clear methodology: The paper clearly describes the de-rendering, simulation, and rendering components.

**Weaknesses:**

W1: Insufficient analysis of fluid domain's fundamental challenges The authors observe performance degradation in the fluids domain but fail to analyze why fluids are fundamentally different from the other tested scenarios. The paper should discuss whether Chain-of-Time is inherently unsuitable for fluid domain due to some specific properties of it like continuous deformation or partial transparency, or whether the issue is specific to their experimental setup. The superficial explanation of "flow rate estimation error" doesn't address why the step-by-step approach that helps with projectile motion actually harms fluid simulation.

W2: Limited temporal horizon evaluation All experiments are constrained to 0.8 seconds of simulation. For a method claiming to improve physical simulation, testing longer time horizons (e.g., 2-5 seconds) would better demonstrate scalability and compound error effects. The mental simulation literature the authors cite often involves longer-term predictions.

W3: Insufficient experimental scope

Model diversity: Only GPT-4o is tested. While the authors mention DALL-E 3's limitations, they should test other recent VLM+IGM combinations (e.g., Gemini Pro Vision, LLaVA variants with diffusion models, or Claude with image generation capabilities).
Domain limitations: The four domains use overly simplistic objects and backgrounds (solid white, uniform blue). The authors should evaluate on some of the established video prediction datasets (Moving MNIST, KTH Actions, BAIR Robot Pushing) or physical reasoning benchmarks (IntPhys, CATER, Physion) to test generalization beyond synthetic scenes, whichever the authors think is suitable for validating their method.

W4: Incomplete related work The related work section misses important connections:

Video prediction literature (e.g., stochastic video generation, physics-informed neural networks)
World models that perform similar step-by-step physical prediction

W5: Sample sizes vary across domains (N=5 to N=20) without justification.

**Questions:**

1. What fundamental properties of fluid simulation make Chain-of-Time perform worse than direct prediction?

2. Can you provide results for simulations beyond 0.8 seconds to assess error accumulation?

3. Have you tested on any established video prediction or physical reasoning benchmarks?

4. Could you test additional VLM+IGM combinations to verify generalizability?

5. Why do sample sizes differ across domains?

---

### Official Review · Reviewer_nzrx · 2025-10-31

**Soundness:** 3
**Presentation:** 3
**Contribution:** 3
**Rating:** 4
**Confidence:** 4

**Summary:**

The paper proposes a way for Image Generation Models to generate k steps in the future for a given input video. The challenge is getting the physics correct when simulating the next k steps. The proposed method is inspired by human mental simulation (where a trajectory is simulated in mind to predict the future state) and Chain of Thought prompting in LLMs (where the model is asked to answer step-by-step). The method works by giving the model input frames and asking the model to simulate small time-steps. Implicitly, the model de-renders, simulates the next step using transition dynamics, and re-renders to give the next frame. They find the physics of the trajectory stay consistent with a small delta t.

**Strengths:**

- shows strong results on 2d motion and gravity scene
- There is a partial success in more complex simulations, like fluids and a bouncing ball
- The method works at inference time and works with existing models like GPT-4o

**Weaknesses:**

- The mechanism is implicit (de-render, transition based on world transition matrix, rendering) and difficult to test
- In 3D scenes, the early error seems to compound, making it difficult to simulate longer time-steps
- Not much comparison with other existing methods (Video or World-Models) for generating physically plausible images generation
- The generalization seems limited to very simple scenes and breaks when applied to more complex physics problems (fluid, bouncing)

**Questions:**

- The number of intuitive physics studies comparing machine learning models and humans' surprise rating to the plausibility of the scene. Are the authors planning on exploring this avenue to see if the mental simulation hypothesis still holds?
- Have you considered testing whether Chain-of-Time generalizes beyond intuitive physics to intuitive psychology? (model vs human rating for plausibility rating of psychology scenes)
- Have the authors explored combining their method with an external physics engine?

---

### Official Review · Reviewer_dVtH · 2025-10-31

**Soundness:** 2
**Presentation:** 3
**Contribution:** 2
**Rating:** 2
**Confidence:** 4

**Summary:**

This paper presents a way to use language models with image generation abilities as a way to simulate physical systems.
The model is presented with a few frames of a physical system (say, a moving ball) with corresponding time steps, and is asked to produce a prediction of the system in the future (up to a specific time) in the form of an image. The paper suggests that doing so "step by step" like chain of thought instead of in a single go produces better results.
The method is demonstrated on a few simple physical scenarios in 2D and 3D.

**Strengths:**

The paper is interesting in its approach and the general context of the problem is important.
It's a well written paper and is easy to follow. I also enjoyed the clarity of the method description and the ample detail given.
I thought using computer vision algorithms to extract the state for better analysis was a nice idea, but see below.

**Weaknesses:**

I think the main issue of the paper is its scope and especially the experimental setup.

I understand why using such simple physical systems was necessary if exact state estimates are needed, but this is a major hinderance for the paper. The experiments only cover a very simple set of physical systems under ideal observation conditions - I feel that to conclude anything about model's abilities to reason about physics, much more detailed and elaborate systems should be examined and analyzed. At this scope of experiments this feels more like an initial study rather than a fully formed paper.

Another weakness of the experiments is that it's not clear if other models would exhibit similar results - using only a single model makes drawing any kind of conclusions about the proposed method quite speculative.

In summary - the paper requires more scope to prove to be a significant contribution to the community.

**Questions:**

above.

---

### Official Review · Reviewer_qjNx · 2025-11-01

**Soundness:** 1
**Presentation:** 2
**Contribution:** 1
**Rating:** 2
**Confidence:** 2

**Summary:**

This paper proposes chain of time simulation, or generating intermediate images during a simulation, to evaluate the capabilities of image generation models on predicting simulated and natural evaluations. They find that this significantly improves performance over the baseline, and suggests that image generation models are able to simulate some properties over time.

**Strengths:**

- Studies an unique topic - physics understanding in image generation models
- Motivates the study from an interdisciplinary perspective

**Weaknesses:**

- limited evaluation. image generation models span a variety of designs, and evaluating only one is insufficient.
- given that gpt's image generation is a closed model with little public detail, it may be difficult to act on these findings to improve image generation models
- limited context from related work
- experimental results are unclear. for example, the paper mentions that figure 6 shows that IGM is able to simulate the projectile's motion because it is close to ground truth, but the pattern does not seem to behave in a way consistent with a physics equation.

**Questions:**

How would you differentiate your method from the visual chain of thought line of work (which also produce intermediate images)? The related work mentions the final goal being that of producing an image, but there is existing literature applying CoT to image generation as well.

Could you provide in the appendix additional images from the evaluation and across time steps? Image generation models often make other mistakes that are not related to physics, which may affect evaluation.

---

### Meta-Review · Area_Chair_fiYp · 2026-01-14

**Summary:**

The paper proposes Chain-of-Time, a method inspired by human mental simulation and in-context reasoning, to enhance the physical simulation capabilities of image generation models. Reviewers found the problem setting to be interesting and appreciated the cognitively inspired motivation, noting that the idea is novel and well-motivated.

However, the reviewers ultimately converged on a rejection decision (scores of 2, 2, 4, and 4), driven by substantial concerns regarding the experimental scope, generalizability, and clarity of the evaluation. Specifically, all reviewers raised concerns about the limited experimental coverage: the method is evaluated only on GPT-4o, making the generalization claims speculative. In addition, the evaluation focuses on overly simplified test settings, including synthetic scenes and very short time horizons (0.8 seconds), and lacks comprehensive comparisons with established baselines and closely related prior work. Some reviewers also questioned the validity of the reported physical reasoning improvements and whether the observed gains are robust or specific to a particular model.

Overall, the limited diversity of evaluated models and the narrow evaluation depth suggest that the contribution remains preliminary. Based on these considerations, the paper does not yet meet the bar for acceptance, and I therefore recommend rejection.

**Reviewer Concerns:**

Since the authors did not submit a rebuttal, none of the reviewers’ concerns or questions were addressed, and all issues remain outstanding.

**Reviewer Scores:**

For each review, all reviewers would likely have kept their original scores unchanged, as there was no rebuttal to inform further discussion or reconsideration.

---

### Decision · Program_Chairs · 2026-01-26

Reject